# The GECo algorithm for Graph Neural Networks Explanation

## Abstract

Graph Neural Networks (GNNs) are powerful models that can manage complex data sources and their interconnection links. One of GNNs' main drawbacks is their lack of interpretability, which limits their application in sensitive fields. In this paper, we introduce a new methodology involving graph communities to address the interpretability of graph classification problems. The proposed method, called GECo, exploits the idea that if a community is a subset of graph nodes densely connected, this property should play a role in graph classification. This is reasonable, especially if we consider the message-passing mechanism, which is the basic mechanism of GNNs. GECo analyzes the contribution to the classification result of the communities in the graph, building a mask that highlights graph-relevant structures. GECo is tested for Graph Convolutional Networks on six artificial and four real-world graph datasets and is compared to the main explainability methods such as PGMExplainer, PGExplainer, GNNExplainer, and SubgraphX using four different metrics. The obtained results outperform the other methods for artificial graph datasets and most real-world datasets.

## 1 Introduction

Deep Neural Networks (DNN) have demonstrated the ability to learn from a wide variety of data, including text, images, and temporal series. However, in cases where information is organized in more complex ways, with individual pieced of data connected by relationships, graph become the preferred data structure. Graphs effectively represent both information elements and their interconnections. This kind of data is commonly found in social network analysis, bioinformatics, chemistry, and finance, where the relationships among data are not less important than the data itself. The synergy between graph data structure and deep neural networks is expressed in graph neural networks (GNNs). These networks combine the flexibility of neural networks with the ability to manage graph-structured data, making it possible to process graph-like data using machine learning methods.

As with other machine learning (ML) models, the ability to provide clear and understandable insights into the reasons behind predictions or decisions is a crucial feature. This explainability is particularly important for critical applications such as medicine, finance, or security.

For neural networks in general, explainability is an open challenge. This paper proposes a new algorithm called GECo (Graph Explainability by COmmunities) to address this challenge and enhance the explainability of GNNs. It first uses the model to classify the entire graph. Then, it detects the different communities, for each community, a smaller subgraph is created, and the model is run to see how likely the subgraph alone supports the predicted class. After evaluating all the communities, an average probability is calculated and set as a threshold. Finally, any community with a probability value higher than the threshold is assessed as necessary for the model's decision. The collection of these key communities forms the final explanation, and from them, the most relevant parts of the graph leading to the classification can be highlighted. We tested the proposed approach's effectiveness on synthetic and real-world datasets, comparing the results with state-of-the-art methodologies and a random baseline. The remainder of the paper is organized in the following way: in Section 2, a review of works in the same field is described; in Section 3, the Graph Neural Networks are described, and the proposed solution is presented in detail, in the same section the used dataset,

along with the evaluation criteria, are described. The results are presented in  Section 4, and the conclusions are drawn in  Section 5.

## 2   RELATED WORKS

The representation power of graphs makes them suitable to describe many real-world data. Citation networks, social networks, chemical molecules, and financial data are directly represented with a graph.  Graph Neural Networks (GNNs) have been conceived to integrate the graphs with the computation capability of neural architectures. They are a robust framework that implements deep learning on graph-related data. Some computation examples are node classification, graph classification, and link predictions. Some of the most popular GNNs, like Graph Convolutional Networks (GCN) (Kipf & Welling, 2017), Graph Attention Networks (GAT) (Veličković et al., 2018), and GraphSage (Hamilton et al., 2017), recursively pass neural messages along the graph edges using the node features and the graph topology information. Using this kind of information leads to complex models; thus, explaining the prediction made by the neural network is challenging. Graph data, unlike images and text, can be less intuitive due to their non-grid structure. The topology of graph data is represented using node features and adjacency matrices, making it less visually apparent than grid-like formats.  Furthermore, each graph node has a different set of neighbours. For these reasons, the traditional explainability methods used for text and images are unsuitable for obtaining convincing explanations for neural graph computation. To effectively explain predictions made by a GNN, it is crucial to identify the most critical input edges, the most important input nodes, the most significant node features, and the input graph that maximizes the prediction for a specific class. Explainability methods for GNNs can be divided into two main groups based on the type of information they provide (Yuan et al., 2022). The first group is referred to as instance-level methods, which focus on explaining predictions by identifying the important input features. These include (1) gradient/features-based methods, which use gradient values or hidden features to approximate input importance and explain predictions through backpropagation; (2) perturbation-based methods, which assess the importance of nodes, edges, or features by evaluating how perturbations in the input affect output; (3) surrogate methods, which involve training more interpretable models on the local neighbourhood of an input node; and (4) decomposition methods, which break down predictions into terms representing the importance of corresponding input features. The second group is referred to as Model-level methods, and study the input graph patterns that lead to specific predictions. Recent literature shows that the state-of-the-art techniques for explainability are the ones we are describing in the following. PGExplainer Luo et al. (2020) and GNNExplainer (Ying et al., 2019) rely on perturbation-based approaches that learn edge masks to highlight important graph components. PG-Explainer trains a mask predictor to estimate edge selection probability, while GNNExplainer refines soft masks for nodes and edges to maximize mutual information between the original and perturbed graph predictions. Xie et al in Xie et al. (2022) propose a technique very close to PGExplainer, that can produce explanations for GNN not tailored to a specific task. It is independent from the specific models and is trained with self-supervision approaches. TAGE is composed of an embedding explainer and a downstream explainer. The first is trained with conditioned contrastive learning, the latter is based on gradient-based explainers. SubgraphX (Yuan et al., 2021) explores subgraph explanations using Monte Carlo Tree Search (MCTS) and Shapley values to find critical subgraphs. Additionally, PGMExplainer (Vu & Thai, 2020) adopts a surrogate approach by constructing a probabilistic graphical model to explain predictions, using perturbations and a Bayesian network to identify important node features.  Despite their effectiveness, all the state-of-the-art methodologies are natively huge because they are based on the combination of several modules, each of a consistent complexity, such as simulators or predictors. This affects their running time, as demonstrated by the experiments carried out in this paper. It has been recently noticed that graph community structure in biological Knowledge Graphs could provide a better grasp of the decision-making of Graph Neural Networks Martínez Mora et al. (2024) The authors of this paper leverage this concept using the game theory as a theoretical supporting idea, considering communities as players of the game focused on maximising the output of the GNN. The method is described as a general framework, but the results are obtained only for node or edge classification in the realm of the Biological Knowledge Graphs. The method we are going to propose here is based on the same theoretical considerations about communities.  However, it considers the whole graph classification and leverages the assumption that communities, with their dense connections, can strongly influence the output of the GNN.

## 3 Materials and Methods

The processing mechanism of the Graph Neural Networks will be introduced in Subsection 3.1, and then the GECo methodology will be described in Subsection 3.2. The datasets used for testing, both synthetic and real, and the parameters calculated for the performance measurements are described in Subsections A.1, A.2, and 3.3, respectively.

### 3.1 Graph Neural Networks

Graph Neural Networks (GNNs) are a particular type of Artificial Neural Network (ANN) used to process data with a graph structure. These models can perform various tasks, such as node classification, graph classification, and link prediction. We are interested in graph classification, where the input is a set of graphs, each belonging to a specific class, and the goal is to predict the class of a given input graph. Let $\mathcal{G} = (\mathcal{V}, \mathcal{E})$ be a graph where $\mathcal{V}$ is the set of nodes and $\mathcal{E}$ is the set of edges. Every graph can be associated with a square matrix called an adjacency matrix, defined as $A \in \mathbb{R}^{|\mathcal{V}| \times |\mathcal{V}|}$. For unweighted graphs $A_{ij} = 1$ if $(i, j) \in \mathcal{E}$, $A_{ij} = 0$ if $(i, j) \notin \mathcal{E}$. For weighted graphs, $A_{ij} = w_{ij}$, but we restrict our studies only to unweighted ones. Every graph node is associated with a vector of features $x \in \mathbb{R}^C$ where $C$ represents the number of features. The set of all node features can be represented using a matrix $X \in \mathbb{R}^{|\mathcal{V}| \times C}$. A GNN network constituted by $K$ layers $l_k$ with $k = 1, \ldots, K$ aims to learn a new matrix representation $H^K \in \mathbb{R}^{|\mathcal{V}| \times F^K}$, where $F^K$ is the number of features per node after processing in layer $K$, exploiting the graph topology information and the node attributes. The idea behind the computation is to update the node representations $H_i^k$ iteratively, combining them with node representations of their neighbours $H_j^k$ with $j \in \mathcal{N}(i)$ (Xu et al., 2019):

$$H_i^{k+1} = UPDATE^k \left( H_i^k, AGGREGATE^k \left( \{ H_j^k, \forall \, j \in \mathcal{N}(i) \} \right) \right) \tag{1}$$

where $UPDATE$ and $AGGREGATE$ are arbitrary differentiable functions and $\mathcal{N}(i)$ represents the set of neighbours of the node $i$. At each iteration, the single node aggregates the information of neighbourhoods, and as the iteration proceeds, each node embedding accumulates information from increasingly distant parts of the graph. This information can be of two kinds: one connected with the structure of the graph, which can be useful in distinguishing structural motifs, and another connected with the features of the nodes in the surroundings. After $K$ iterations, the computed node embeddings are affected by the features of nodes that are $K$-hops away.

### 3.2 The proposed methodology

The proposed method aims to find subgraphs responsible for a large part of the output value of the Graph Neural Network. The algorithm is based on the hypothesis that a GNN will learn to recognize some structures in the input graph; subgraphs with these structures will produce a high response in output. Communities are structures easily recognized in a graph; intuitively, a community is a subset of nodes whose connections among each are denser than those with the rest of the network. Considering the **aggregate** step in the GNN algorithm already discussed, relevant communities should generate a very high quote of the output value in a trained neural network. The community subgraphs are identified and proposed as stand-alone subgraphs to the GNN, and the different output values are memorized using the proposed method. The subgraph corresponding to the highest output values is considered the most important for the classification output.

Regarding the taxonomy described in Section 2, the method presented here belongs to the instance-level methods, particularly the perturbation-based methods, because it identifies a perturbation of the input consistent with the prediction of the GNN on the original graph.

Going into detail, given a trained GNN $f$, the proposed method aims to find a mask containing the most relevant nodes for the final classification. The algorithm's inputs are the trained GNN $f$, a graph $\mathcal{G}$ and the associated label $y$. In the first step, the graph $\mathcal{G}$ is given in input to the GNN, obtaining the prediction $\hat{y}$ (see Figure 1a).

In the second step, it is necessary to find the communities of the graph; we decided to use a community detection greedy algorithm based on modularity. Community detection is a well-studied problem in graph theory. The goal is to find groups of nodes that are more similar to each other

than to other nodes. Several algorithms exist to solve the community detection problem. Girvan and Newman introduced the most popular algorithm in 2002 based on the computation of the edge betweenness Girvan & Newman (2002). There are other methods based on modularity optimization. Modularity is a measure that quantifies the density of connections within a community Newman & Girvan (2004). The most popular approaches use greedy algorithms, such as Newman (2004) Newman (2004) and Clauset et al. (2004) Clauset et al. (2004). Another popular approach based on modularity optimization, which instead uses a heuristic approach, is the Louvain method introduced by Blondel et al. in 2008 Blondel et al. (2008). Our approach uses the algorithm proposed by Clauset et al. Clauset et al. (2004) based on a greedy algorithm. We chose this algorithm because it works well even for large graphs and uses data structures for sparse matrices, decreasing the algorithm's computational complexity concerning the first implementation of Newman (2004) Newman (2004). The modularity is defined as:

$$Q = \frac{1}{2m} \sum_{ij} \left( A_{ij} - \gamma \frac{k_i k_j}{2m} \right) \delta(c_i, c_j) \tag{2}$$

where $m$ is the number of edges of the graph, $A_{ij}$ is an element of the adjacency matrix $A$ of the graph, $k_i$ and $k_j$ are the nodes degree and $\delta(c_i, c_j) = 1$ if the two nodes belong to the same community, 0 vice versa. The value $\gamma$ is called the resolution parameter, and it is an arbitrary tradeoff between intra-group edges and inter-group edges. It is widespread to use $\gamma = 1$. If $\gamma < 1$, the modularity favors larger communities, and vice versa, smaller ones. The goal is to find the partition that maximizes $Q$. The pseudocode of this algorithm is in Algorithm 1.

---

**Algorithm 1** Community Detection Greedy Algorithm Clauset et al. (2004)

---

**Require:** Graph $\mathcal{G} = (\mathcal{V}, \mathcal{E})$ with $n = |\mathcal{V}|$ nodes
**Ensure:** Set of communities $C$
1: **Step 1:** Initialize each node in its own community, resulting in $n$ communities
2: $C \leftarrow \{\{v\} \mid v \in \mathcal{V}\}$
3: **while** $|C| > 1$ **do**
4:     **Step 2:** Compute the modularity variation $\Delta Q$ for each pair of communities connected by at least one edge
5:     **for** each pair of communities $(C_i, C_j) \in C$ connected by at least one edge **do**
6:         Compute the modularity variation $\Delta Q$ if $C_i$ and $C_j$ are merged
7:     **end for**
8:     **Step 3:** Identify the community pairs $(C_i, C_j)$ for which $\Delta Q$ is the largest and merge them
9:     $C \leftarrow (C \setminus \{C_i, C_j\}) \cup \{C_i \cup C_j\}$
10:     **Step 4:** Record the modularity $Q$ for the current partition
11:     Note that the modularity $Q$ is computed for the whole graph
12: **end while**
13: **Step 5:** Select the partition for which $Q$ is maximal
14: **return** $C$

---

For each community, we build a subgraph that contains only the nodes that belong to the considered community. These subgraphs are fed to the GNN, and the probability value corresponding to the predicted class $\hat{y}$ is stored (see Figure 1c). After this step, we have the associated value of the probability for each community. We use these values to calculate a threshold $\tau$ using, for example, the mean or the median of the probability values. Fixed the value of $\tau$, we consider the communities associated with a probability value greater than $\tau$, and we use the nodes that belong to these communities to form the final explanation (see Figure 1e). .

The algorithm pseudocode of the proposed approach is in Algorithm 2.

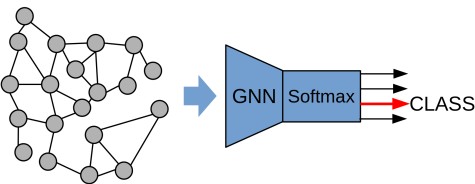

(a) Step 1: classification of the whole graph $\mathcal{G}$ in the class CLASS

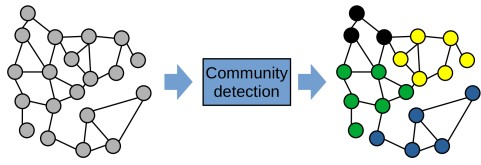

(b) Step 2: Community detection of the input graph $\mathcal{G}$.

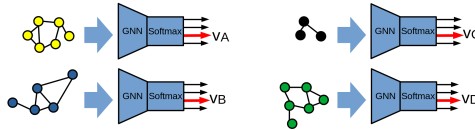

(c) Step 3: Classification of every single community of the input graph $\mathcal{G}$.

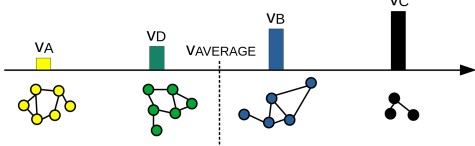

(d) Step 4: Decision on $\tau = V_{AVERAGE}$ threshold value.

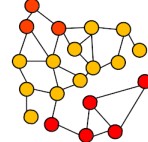

(e) Step 5: Identification of the most influential community of the input graph $\mathcal{G}$.

Figure 1: The steps of the proposed GECo algorithm.

---

**Algorithm 2** GECo algorithm

---

**Require:** Trained GNN model $f$, Test graph $\mathcal{G}$
**Ensure:** Explanation mask $E$
1: **Step 1:** Classify the whole graph $\mathcal{G}$ using $f$ and store the predicted class $\hat{y}$
2: $\hat{y} \leftarrow f(\mathcal{G})$
3: **Step 2:** Identify the communities $\{C\}$ of the graph $\mathcal{G}$
4: $\{C\} \leftarrow \text{CommunityDetection}(\mathcal{G})$
5: **for** each community $C_i$ in $\{C\}$ **do**
6:     **Step 3:** Build the corresponding subgraph $\mathcal{G}_i$ and feed it to $f$
7:     $\mathcal{G}_i \leftarrow \text{Subgraph}(\mathcal{G}, C_i)$
8:     $p_i \leftarrow f(\mathcal{G}_i)[\hat{y}]$
9:     Save the probability value $p_i$
10: **end for**
11: **Step 4:** Compute the threshold value $\tau$ by averaging the probability values $\{p_i\}$
12: $\tau \leftarrow \frac{1}{|\{C\}|} \sum_i p_i$
13: **Step 5:** Select communities with probability values greater than $\tau$ and use their nodes to compose the final explanation mask $E$
14: $E \leftarrow \bigcup_{i:p_i > \tau} C_i$
15: **return** $E$

---

### 3.3 EVALUATION CRITERIA

Several metrics have been introduced to evaluate the effectiveness of a method that explains the result obtained using a GNN. In particular, the considered metrics leverage predicted and ground-truth explanations and use user-controlled parameters such as the probability distribution obtained by the GNN and the number of important features that compose the explanation mask. **Fidelity** (Pope et al., 2019) studies how the prediction of the model changes if we remove from the original graph nodes/edges/node features. Let $\mathcal{G}_i$ the i-th graph of the test set, $y_i$ the true label of the graph, $\hat{y}_i^{\mathcal{G}_i}$ the label predicted by the GNN, $m_i$ the mask produced by the explanation algorithm, $\mathcal{G}_i \setminus m_i$

the graph without the nodes that belong to the mask, $\mathcal{G}_i^{m_i}$ the graph with only the important features detected by the algorithm, $\hat{y}_i^{\mathcal{G}_i \setminus m_i}$ and $\hat{y}_i^{\mathcal{G}_i^{m_i}}$ the labels obtained feeding the GNN with the graphs $\mathcal{G}_i \setminus m_i$ and $\mathcal{G}_i^{m_i}$. It is possible to define two measures $Fid^+$ and $Fid^-$ as:

$$g(y, \hat{y}) = \begin{cases} 1 \ y = \hat{y} \\ 0 \ y \neq \hat{y} \end{cases} \qquad Fid^+ = \frac{1}{N} \sum_{i=1}^{N} \left| g(\hat{y}_i, y_i) - g\left(\hat{y}_i^{\mathcal{G}_i \setminus m_i}, y_i\right) \right| \qquad (3)$$

$$Fid^- = \frac{1}{N} \sum_{i=1}^{N} |g(\hat{y}_i, y_i) - g(\hat{y}_i^{m_i}, y_i)| \qquad (4)$$

where $N$ is the number of graphs in the test set. $Fid^+$ studies how the prediction changes if we remove the essential features (edges/features/nodes) identified by the explanation algorithm from the original graph. High values indicate good explanations, so the features detected by the algorithm are the most discriminative. The $Fid^-$ measure studies how the predictions change if we consider only the features detected by the explanation algorithm. Low values indicate that the algorithm identifies the most discriminative features and does not consider the least ones.

Using the definition of $Fid^+$ and $Fid^-$ measures, it is possible to define two properties that a reasonable explanation should have: necessity and sufficiency (Amara et al., 2022). An explanation is necessary if the model prediction changes if we remove the features belonging to the explanation from the graph. A necessary explanation has a $Fid^+$ close to 1. Conversely, an explanation is sufficient if it leads independently to the model's original prediction. A sufficient explanation has a $Fid^-$ close to 0. The so-called **Characterization Score** (*charact*) is a global evaluation metric that considers $Fid^+$ and $Fid^-$ (Amara et al., 2022). This measure is helpful because it balances explanations' sufficiency and necessity requirements. *charact* is the harmonic mean of $Fid^+$ and $1 - Fid^-$, and it is defined as follows:

$$charact = \frac{w_+ + w_-}{\frac{w_+}{Fid^+} + \frac{w_-}{1 - Fid^-}} \qquad (5)$$

where $w_+, w_- \in [0, 1]$ are respectively the weights for $Fid^+$ and $1 - Fid^-$ and satisfy $w_+ + w_- = 1$. **Graph Explanation Accuracy** (GEA) Agarwal et al. (2023) is an evaluation strategy that measures the correctness of an explanation using the ground-truth explanation $M^g$. Ground truth and predicted explanations are binary vectors where 0 means an attribute is not essential and 1 is important for the model prediction. To measure accuracy, we used the Jaccard index between the ground truth $M^g$ and predicted $M^p$:

$$JAC\left(M^g, M^p\right) = \frac{TP(M^g, M^p)}{TP(M^g, M^p) + FP(M^g, M^p) + FN(M^g, M^p)} \qquad (6)$$

where TP denotes true positives, FP false positives, and FN false negatives. If we define as $\zeta$ the set of all possible ground-truth explanations, where $|\zeta| = 1$ for graphs having a unique explanation. GEA can be calculated as:

$$GEA(\zeta, M^p) = \max\{JAC(M^g, M^p)\} \ \forall \ M^g \in \zeta. \qquad (7)$$

# 4 RESULTS

Our experimental workflow starts with training a simple GNN made of three GCN layers, an average readout layer, and finally, a linear layer whose number of units is equal to the classes of the considered dataset with softmax activation. We performed a set of preliminary experiments to find the configuration and the suitable number of epochs for the best classification accuracy results. For the synthetic dataset, each GCN layer's dimensionality is 20, while it is 64 for real-world ones. We train the GNN using the Adam optimization algorithm (Kingma & Ba, 2015) with a batch size of 64 and a learning rate of 0.05 in both cases. To ensure reliable results, as a validation protocol, we used 100 different random splits in training and testing with a ratio of 8:2. Post-training the GNN, we applied the explanation algorithm on the test set and computed the performance indices described in Subsection 3.3. For the GNN computation, each graph node must be associated with a feature vector: in

the synthetic dataset, we assigned each node a feature vector representing its degree; conversely, in the real-world datasets, each node has a feature vector of dimension 14 that represents the chemical properties of the corresponding atom. For a thorough assessment of our method's performance, we compared it against a random baseline and five state-of-the-art methodologies: PGMExplainer (Vu & Thai, 2020), PGExplainer (Luo et al., 2020), GNNExplainer (Ying et al., 2019), and SubgraphX (Yuan et al., 2021) and Tage (Xie et al., 2022). The random baseline mask on nodes was obtained using a Bernoulli distribution with a 0.5 probability value.

### 4.1 RESULTS ON SYNTHETIC DATASETS

Here, we analyze and discuss the results assessed by the GECo algorithm on synthetic datasets, comparing them with those from a random baseline and the state-of-the-art techniques previously described. In order to test the proposed methodologies, it is necessary to have datasets of graphs with a corresponding mask that highlights the relevant features for classification. This mask allows us to compare the result of the algorithm's explanation with the ground truth explanation. For this reason, we created six synthetic datasets that contain ground truth explanations. The description of these datasets is reported in Appendix A.1. We reported the results and the relative comparison in Table 1 where up-arrows indicate measures that ideally should be one, and down arrows indicate measures that ideally should be zero. From the table, we observe that the GECo algorithm consistently excels in explainability across different datasets, as indicated by its high $Fid^+$ values and near-zero $Fid^-$ values. For instance, in the ba_house_cycle dataset, GECo achieves $Fid^+ = 0.929$ and $Fid^- = 0$, effectively identifying key features while excluding irrelevant ones. This suggests that GECo focuses on the most discriminative features used in decision-making. In contrast, methods like GNNExplainer, despite detecting important features, tend to include irrelevant ones. For example, GNNExplainer reports $Fid^+ = 0.478$ and $Fid^- = 0.257$, indicating a less precise explanation compared to GECo. The *charact* metric further demonstrates GECo's ability to balance sufficiency and necessity, outperforming methods like PGExplainer, which struggles with lower scores due to weaker performance in either aspect. Regarding explanation correctness, as measured by the *GEA* metric, GECo consistently provides reliable and accurate explanations. For instance, it aligns well with ground-truth explanations in synthetic datasets like ba_cycle_wheel, where it correctly detects wheel motifs, as illustrated in Figure 2. We measured explanation time (in seconds) for each test set graph, averaging over 100 splits. GECo consistently outperforms others, with under 3 seconds on average, while SubgraphX takes the longest, over 100 seconds. GNNExplainer and PGExplainer are slower than PGMExplainer, each taking around 100 seconds across datasets. tage results the second fastest with about 7 seconds. These times refer to the ba_cycle_wheel dataset, but the same trend is observable for the other datasets. Overall, GECo stands out as the most consistent and robust explainability method across both binary and multiclass datasets. It maintains an excellent balance between sufficiency and necessity requirements (high $Fid^+$, low $Fid^-$, and high *charact* scores) while closely aligning with ground truth (high *GEA*). This makes GECo the most reliable option for generating clear, concise, and accurate explanations in GNN-based models.

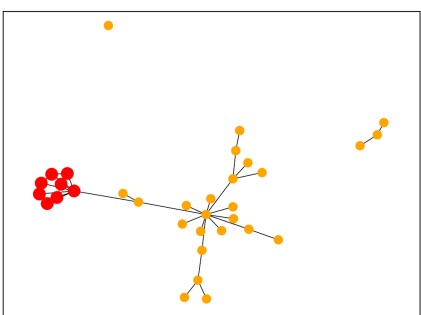

Figure 2: An example of an explanation for a graph belonging to the wheel-motif class. The red nodes are the ones composing the explanation mask.

Table 1: Results on synthetic datasets and comparison of GECo with state-of-the-art methods, using the chosen 4 evaluation metrics. The best results are in bold and report mean±standard deviation over 100 runs.

| Dataset | Method | $Fid^+ \uparrow$ | $Fid^- \downarrow$ | $charact \uparrow$ | $GEA \uparrow$ |
|---|---|---|---|---|---|
| ba_house_cycle | random | $0.411 \pm 0.067$ | $0.410 \pm 0.053$ | $0.477 \pm 0.055$ | $0.145 \pm 0.004$ |
| | PGMExplainer | $0.270 \pm 0.056$ | $0.436 \pm 0.048$ | $0.360 \pm 0.048$ | $0.089 \pm 0.006$ |
| | PGExplainer | $0.217 \pm 0.137$ | $0.459 \pm 0.101$ | $0.292 \pm 0.152$ | $0.213 \pm 0.152$ |
| | GNNExplainer | $0.478 \pm 0.044$ | $0.257 \pm 0.066$ | $0.579 \pm 0.037$ | $0.190 \pm 0.001$ |
| | SubgraphX | $0.191 \pm 0.181$ | $0.356 \pm 0.252$ | $0.270 \pm 0.233$ | $0.269 \pm 0.147$ |
| | tage | $0.426 \pm 0.079$ | $0.269 \pm 0.072$ | $0.533 \pm 0.071$ | $0.132 \pm 0.021$ |
| | GECo | $\mathbf{0.929 \pm 0.043}$ | $\mathbf{0.000 \pm 0.002}$ | $\mathbf{0.952 \pm 0.024}$ | $\mathbf{0.305 \pm 0.029}$ |
| ba_cycle_wheel | random | $0.297 \pm 0.033$ | $0.294 \pm 0.034$ | $0.417 \pm 0.035$ | $0.172 \pm 0.005$ |
| | PGMExplainer | $0.133 \pm 0.024$ | $0.360 \pm 0.040$ | $0.219 \pm 0.034$ | $0.113 \pm 0.007$ |
| | PGExplainer | $0.111 \pm 0.193$ | $0.424 \pm 0.133$ | $0.139 \pm 0.237$ | $0.155 \pm 0.149$ |
| | GNNExplainer | $0.491 \pm 0.035$ | $0.026 \pm 0.018$ | $0.652 \pm 0.032$ | $0.234 \pm 0.003$ |
| | SubgraphX | $0.329 \pm 0.231$ | $0.218 \pm 0.300$ | $0.439 \pm 0.306$ | $0.380 \pm 0.181$ |
| | tage | $0.499 \pm 0.065$ | $0.005 \pm 0.005$ | $0.662 \pm 0.059$ | $0.221 \pm 0.025$ |
| | GECo | $\mathbf{0.607 \pm 0.100}$ | $\mathbf{0.000 \pm 0.000}$ | $\mathbf{0.750 \pm 0.076}$ | $\mathbf{0.553 \pm 0.032}$ |
| er_house_cycle | random | $0.368 \pm 0.041$ | $0.372 \pm 0.040$ | $0.461 \pm 0.030$ | $0.154 \pm 0.005$ |
| | PGMExplainer | $0.203 \pm 0.053$ | $0.430 \pm 0.040$ | $0.295 \pm 0.058$ | $0.097 \pm 0.006$ |
| | PGExplainer | $0.286 \pm 0.154$ | $0.426 \pm 0.141$ | $0.367 \pm 0.169$ | $0.265 \pm 0.174$ |
| | GNNExplainer | $0.471 \pm 0.039$ | $0.084 \pm 0.030$ | $0.621 \pm 0.035$ | $0.197 \pm 0.021$ |
| | SubgraphX | $0.260 \pm 0.111$ | $0.349 \pm 0.110$ | $0.363 \pm 0.132$ | $0.262 \pm 0.130$ |
| | tage | $0.435 \pm 0.087$ | $0.317 \pm 0.148$ | $0.526 \pm 0.093$ | $0.122 \pm 0.036$ |
| | GECo | $\mathbf{0.791 \pm 0.090}$ | $\mathbf{0.000 \pm 0.001}$ | $\mathbf{0.880 \pm 0.057}$ | $\mathbf{0.391 \pm 0.069}$ |
| er_cycle_wheel | random | $0.348 \pm 0.055$ | $0.352 \pm 0.058$ | $0.448 \pm 0.038$ | $0.170 \pm 0.005$ |
| | PGMExplainer | $0.213 \pm 0.057$ | $0.442 \pm 0.046$ | $0.303 \pm 0.056$ | $0.113 \pm 0.005$ |
| | PGExplainer | $0.222 \pm 0.164$ | $0.414 \pm 0.134$ | $0.295 \pm 0.190$ | $0.227 \pm 0.167$ |
| | GNNExplainer | $0.454 \pm 0.044$ | $0.048 \pm 0.036$ | $0.613 \pm 0.040$ | $0.227 \pm 0.003$ |
| | SubgraphX | $0.167 \pm 0.124$ | $0.392 \pm 0.133$ | $0.243 \pm 0.160$ | $0.263 \pm 0.135$ |
| | tage | $0.466 \pm 0.097$ | $0.164 \pm 0.151$ | $0.594 \pm 0.109$ | $0.208 \pm 0.045$ |
| | GECo | $\mathbf{0.866 \pm 0.108}$ | $\mathbf{0.002 \pm 0.018}$ | $\mathbf{0.923 \pm 0.070}$ | $\mathbf{0.407 \pm 0.050}$ |
| ba_cycle_wheel_grid | random | $0.467 \pm 0.034$ | $0.468 \pm 0.036$ | $0.495 \pm 0.023$ | $0.185 \pm 0.004$ |
| | PGMExplainer | $0.229 \pm 0.035$ | $0.127 \pm 0.005$ | $0.299 \pm 0.030$ | $0.127 \pm 0.005$ |
| | PGExplainer | $0.214 \pm 0.172$ | $0.568 \pm 0.099$ | $0.260 \pm 0.168$ | $0.159 \pm 0.114$ |
| | GNNExplainer | $0.664 \pm 0.032$ | $0.147 \pm 0.038$ | $0.746 \pm 0.026$ | $0.256 \pm 0.003$ |
| | SubgraphX | $0.562 \pm 0.142$ | $0.217 \pm 0.197$ | $0.650 \pm 0.161$ | $0.527 \pm 0.115$ |
| | tage | $0.644 \pm 0.074$ | $0.111 \pm 0.058$ | $0.744 \pm 0.057$ | $0.236 \pm 0.024$ |
| | GECo | $\mathbf{0.887 \pm 0.052}$ | $\mathbf{0.000 \pm 0.002}$ | $\mathbf{0.939 \pm 0.029}$ | $\mathbf{0.561 \pm 0.036}$ |
| er_cycle_wheel_grid | random | $0.520 \pm 0.039$ | $0.521 \pm 0.034$ | $0.496 \pm 0.016$ | $0.185 \pm 0.004$ |
| | PGMExplainer | $0.331 \pm 0.051$ | $0.623 \pm 0.036$ | $0.349 \pm 0.032$ | $0.128 \pm 0.005$ |
| | PGExplainer | $0.285 \pm 0.153$ | $0.557 \pm 0.135$ | $0.333 \pm 0.151$ | $0.205 \pm 0.143$ |
| | GNNExplainer | $0.628 \pm 0.041$ | $0.074 \pm 0.024$ | $0.748 \pm 0.033$ | $0.246 \pm 0.002$ |
| | SubgraphX | $0.325 \pm 0.087$ | $0.410 \pm 0.095$ | $0.415 \pm 0.089$ | $0.378 \pm 0.098$ |
| | tage | $0.648 \pm 0.073$ | $0.091 \pm 0.051$ | $0.754 \pm 0.061$ | $0.256 \pm 0.014$ |
| | GECo | $\mathbf{0.915 \pm 0.028}$ | $\mathbf{0.001 \pm 0.000}$ | $\mathbf{0.954 \pm 0.017}$ | $\mathbf{0.510 \pm 0.038}$ |

## 4.2 Results on real-world data

In the following, we analyze and discuss the results achieved by the GECo algorithm on real-world datasets, comparing them with random baseline and state-of-the-art techniques. In our experimental activity we have considered a containing molecules since we can obtain the ground-truth explanations for these graphs and compare them with those obtained by our proposal. The description of these datasets is reported in Abbendix A.2. It is essential to point out that the ground-truth explanation is only available for the positive class in the molecular datasets used in our experimental activity. For instance, regarding the Benzene dataset, a molecule belonging to the positive class contains a benzene ring, so the ground-truth explanation comprises the nodes corresponding to the atoms composing the benzene ring. Conversely, molecules of the negative class do not include the benzene ring, so the ground-truth explanation does not contain any node. However, all the explanation algorithms always respond by explaining net response, providing an explanation mask since they try to find the most relevant features used by the model during the decision-making process. As a result, when calculating the *GEA* metric (see Equation 7), a graph belonging to the negative class always yields a *GEA* of 0 since $TP = 0$ and leading to low overall *GEA* values. Furthermore, to avoid division by zero, when $TP + FP + FN = 0$, we add a small constant $\epsilon = 1 \times 10^{-9}$ to the denominator. Table 2 reports the results assessed by the GECo algorithm and the relative comparison with state-of-the-art approaches; the up and down arrows have the same meaning in Table 1. GECo demonstrates superior performance in identifying highly sufficient features, as indicated by its $Fid^+$ values, where it outperforms other approaches in all datasets except for the Mutagenicity dataset. This indicates that the features selected by GECo can nearly perfectly predict the model's original output. In terms of $Fid^-$ values, GECo also excels, suggesting it can effectively identify sufficient features for accurate predictions independently. This trend is reinforced by the *charact* metric, where GECo achieves the highest scores, reflecting its excellent balance of necessary and sufficient requirements across most datasets. In real-world applications, it is essential to ensure that the features identified by an explanation algorithm correspond to those recognized by domain experts, and this is evaluated using the *GEA* metric. GECo outperforms other methods in this regard, aligning its predicted explanations closely with ground-truth explanations. For instance, in the Mutagenicity dataset, illustrated in Figure 3, GECo correctly identifies the atoms associated with the amino group $NH_2$, along with additional hydrogen and calcium atoms. In terms of explanation time for this kind of dataset, we observe that GECo is the fastest algorithm, taking around 18 seconds. PGMExplainer and tage are the second fastest, at around 130 seconds. PGExplainer, GNNExplainer, and SubgraphX take significantly longer, over 700 seconds. These values refer to the Benzene dataset, but we observed the same trend for the other datasets. Overall, GECo performs remarkably on real-world datasets compared to random baselines and other state-of-the-art approaches. It effectively detects highly sufficient features that can predict the model's output while discarding irrelevant ones, achieving an optimal balance between necessity and sufficiency. Moreover, GECo's explanations align well with ground truth, although some minor discrepancies in functional group localization suggest potential areas for improvement. Its generalization ability is commendable, indicating room for further refinement in future works.

## 5 Conclusions

This paper introduces GECo, a novel GNN explainability methodology that leverages community detection for graph classification tasks. By focusing on community subgraphs, it identifies key graph structures crucial for the decision-making process. Through extensive experimentation on both synthetic and real-world datasets, GECo consistently outperformed state-of-the-art approaches. It demonstrated superior performance in detecting relevant features and meeting sufficiency (low values of $Fid^-$) and necessity requirements (high values of $Fid^+$) while also aligning well with ground-truth explanations. Moreover, its computational efficiency makes it a practical solution for large-scale applications. Future work will aim to further refine GECo, particularly in enhancing feature localization in complex datasets and studying its sensibility to the community detection algorithm.

Table 2: Results real-world datasets. The legend is as the one reported in Table 1

| Dataset | Method | $Fid^+ \uparrow$ | $Fid^- \downarrow$ | $charact \uparrow$ | $GEA \uparrow$ |
|---|---|---|---|---|---|
| Mutagenicity | random | $0.561 \pm 0.112$ | $0.563 \pm 0.111$ | $0.468 \pm 0.030$ | $0.033 \pm 0.003$ |
| | PGMExplainer | $0.464 \pm 0.128$ | $0.600 \pm 0.076$ | $0.408 \pm 0.051$ | $0.031 \pm 0.004$ |
| | PGExplainer | $0.244 \pm 0.137$ | $0.311 \pm 0.106$ | $0.344 \pm 0.160$ | $0.038 \pm 0.049$ |
| | GNNExplainer | $\mathbf{0.640 \pm 0.066}$ | $0.086 \pm 0.016$ | $\mathbf{0.750 \pm 0.054}$ | $0.033 \pm 0.003$ |
| | SubgraphX | $0.216 \pm 0.119$ | $0.275 \pm 0.076$ | $0.319 \pm 0.140$ | $0.025 \pm 0.034$ |
| | tage | $0.356 \pm 0.078$ | $0.357 \pm 0.078$ | $0.451 \pm 0.063$ | $0.054 \pm 0.006$ |
| | GECo | $0.481 \pm 0.058$ | $\mathbf{0.004 \pm 0.003}$ | $0.647 \pm 0.051$ | $\mathbf{0.111 \pm 0.011}$ |
| Benzene | random | $0.437 \pm 0.023$ | $0.436 \pm 0.023$ | $0.491 \pm 0.009$ | $0.111 \pm 0.002$ |
| | PGMExplainer | $0.219 \pm 0.045$ | $0.243 \pm 0.048$ | $0.336 \pm 0.053$ | $0.085 \pm 0.002$ |
| | PGExplainer | $0.211 \pm 0.053$ | $0.504 \pm 0.046$ | $0.292 \pm 0.052$ | $0.048 \pm 0.041$ |
| | GNNExplainer | $0.499 \pm 0.016$ | $0.106 \pm 0.009$ | $0.640 \pm 0.013$ | $0.143 \pm 0.002$ |
| | SubgraphX | $0.290 \pm 0.071$ | $0.274 \pm 0.085$ | $0.407 \pm 0.074$ | $0.133 \pm 0.081$ |
| | tage | $0.507 \pm 0.037$ | $0.132 \pm 0.053$ | $0.639 \pm 0.041$ | $0.184 \pm 0.009$ |
| | GECo | $\mathbf{0.710 \pm 0.053}$ | $\mathbf{0.015 \pm 0.010}$ | $\mathbf{0.824 \pm 0.039}$ | $\mathbf{0.236 \pm 0.018}$ |
| Fluoride-Carbonyl | random | $0.173 \pm 0.070$ | $0.172 \pm 0.069$ | $0.276 \pm 0.088$ | $0.034 \pm 0.001$ |
| | PGMExplainer | $0.066 \pm 0.011$ | $0.099 \pm 0.027$ | $0.123 \pm 0.069$ | $0.023 \pm 0.001$ |
| | PGExplainer | $0.129 \pm 0.042$ | $0.288 \pm 0.092$ | $0.215 \pm 0.060$ | $\mathbf{0.058 \pm 0.025}$ |
| | GNNExplainer | $0.193 \pm 0.096$ | $0.084 \pm 0.020$ | $0.309 \pm 0.121$ | $0.039 \pm 0.002$ |
| | SubgraphX | $0.133 \pm 0.040$ | $0.162 \pm 0.081$ | $0.227 \pm 0.060$ | $0.020 \pm 0.009$ |
| | tage | $0.588 \pm 0.094$ | $0.332 \pm 0.150$ | $0.606 \pm 0.077$ | $0.046 \pm 0.003$ |
| | GECo | $\mathbf{0.615 \pm 0.083}$ | $\mathbf{0.021 \pm 0.008}$ | $\mathbf{0.751 \pm 0.065}$ | $0.038 \pm 0.005$ |
| Alkane-Carbonyl | random | $0.253 \pm 0.039$ | $0.255 \pm 0.037$ | $0.375 \pm 0.041$ | $0.041 \pm 0.005$ |
| | PGMExplainer | $0.094 \pm 0.024$ | $0.299 \pm 0.051$ | $0.165 \pm 0.037$ | $0.027 \pm 0.005$ |
| | PGExplainer | $0.090 \pm 0.080$ | $0.382 \pm 0.062$ | $0.146 \pm 0.106$ | $0.045 \pm 0.021$ |
| | GNNExplainer | $0.304 \pm 0.051$ | $0.092 \pm 0.022$ | $0.454 \pm 0.057$ | $0.041 \pm 0.003$ |
| | SubgraphX | $0.188 \pm 0.081$ | $0.149 \pm 0.094$ | $0.302 \pm 0.114$ | $0.041 \pm 0.019$ |
| | tage | $0.325 \pm 0.089$ | $0.279 \pm 0.081$ | $0.435 \pm 0.071$ | $0.044 \pm 0.013$ |
| | GECo | $\mathbf{0.575 \pm 0.046}$ | $\mathbf{0.001 \pm 0.003}$ | $\mathbf{0.728 \pm 0.038}$ | $\mathbf{0.066 \pm 0.009}$ |

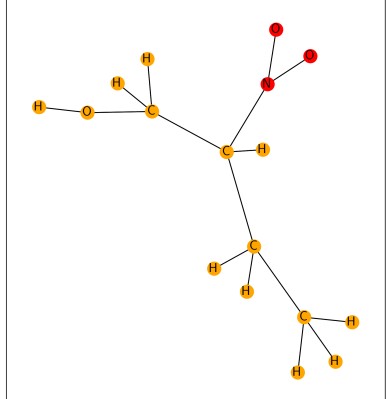 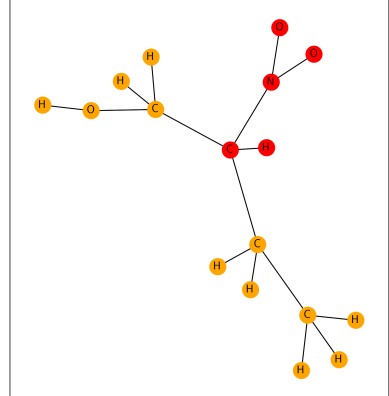

Figure 3: Explanation example for a graph belonging to the Mutagenicity dataset. On the left, we have the ground-truth explanation; on the right, we have the predicted mask. In either case, the red nodes are the ones composing the mask.

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

## A   APPENDIX

### A.1   SYNTHETIC DATASETS

We considered the following graphs to generate the synthetic datasets: Erdös-Rényi (ER) and Barabasi-Albert (BA). ER graphs (Erdös & Rényi, 1959; Gilbert, 1959) are random graphs introduced by Erdös-Rényi in 1959. Generally, an ER graph has a fixed number of nodes $n$ connected by randomly created edges. There are two main models: $G(n, p)$, where each edge is added to the graph with probability $p$, and the $G(n, M)$ model, where a fixed number $M$ of edges are chosen uniformly at random from all possible edges. BA model (Barabási & Albert, 1999) is an algorithm to generate a random scale-free network using the technique of "preferential attachment". A random scale-free network is characterized by a degree distribution that follows a power-law. In such a network, most nodes have only a few connections (low degree), while a few nodes (called hubs) have many connections (high degree). This contrasts with random networks where the degree distribution is more uniform, and most nodes have a similar number of connections. During the creation of the synthetic datasets, we added to the graphs one of the motifs depicted in Figure 4. Using the ER and BA graphs and the after-mentioned motifs, we built the following synthetic datasets:

- **ba_house_cycle**: contains 1000 BA graphs with 25 nodes. We attach to 500 graphs a house motif and to the other 500 a cycle 6 motif.

- **er_house_cycle**: contains 1000 ER graphs with 25 nodes. We attach a house motif to 500 graphs and a cycle 6 motif to the other 500 graphs.

- **ba_cycle_wheel**: contains 1000 BA graphs with 25 nodes. We attach to 500 graphs a cycle 5 motif and a wheel motif to the remaining 500 graphs.

- **er_cycle_wheel**: contains 1000 ER graphs with 25 nodes. We attach to 500 graphs a cycle 5 motif and a wheel motif to the remaining 500 graphs.

- **ba_cycle_wheel_grid**: contains 1500 BA graphs with 25 nodes. We attach to 500 graphs a cycle 5 motif, a wheel motif to another 500 graphs, and to the remaining part a grid motif.

- **er_cycle_wheel_grid**: contains 1500 ER graphs with 25 nodes. We attach a cycle 5 motif to 500 graphs, a wheel motif to another 500 graphs, and to the remaining part a grid motif.

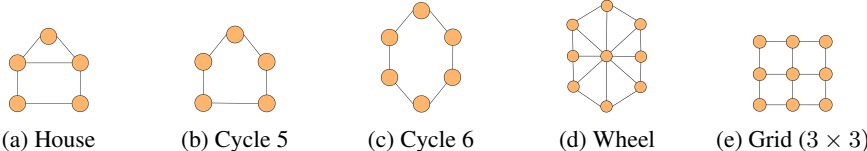

(a) House     (b) Cycle 5     (c) Cycle 6     (d) Wheel     (e) Grid $(3 \times 3)$

Figure 4: Motifs used in synthetic datasets.

## A.2 REAL-WORLD DATASETS

The datasets involved in our experimental activity are the following:

- *Mutagenicity* (Kazius et al., 2005): it contains 1768 molecular graphs labelled into two classes according to their mutagenicity properties. The graph labels correspond to the presence or absence of toxicophores: $NH_2$, $NO_2$, aliphatic halide, nitroso, and azo-type.

- *Benzene* (Sanchez-Lengeling et al., 2020): it consists of 12000 molecular graphs from the ZINC15 database (Sterling & Irwin, 2015). The molecules belong to two classes, and the goal is to predict whether a given molecule contains a benzene ring. For this dataset, the ground-truth explanations are the atoms (nodes) composing the benzene ring, and if the molecule contains multiple benzene rings, each of these forms the explanation.

- *Fluoride-Carbonyl* (Sanchez-Lengeling et al., 2020): it contains 8761 molecular graphs labelled according to two classes. For this dataset, a positive sample indicates that the molecule comprises a fluoride ($F^-$) and carbonyl ($C = O$) functional group, so the ground-true explanations consist of combinations of fluoride atoms and carbonyl functional groups.

- *Alkane-Carbonyl* (Sanchez-Lengeling et al., 2020): it contains 1125 molecular graphs labelled according to two classes. A positive sample represents a molecule containing an unbranched alkane and a carbonyl ($C = O$) functional group, so the ground-truth explanations include combinations of alkane and carbonyl functional groups.

