# OpenReview forum: "The GECo algorithm for Graph Neural Networks Explanation"
_ICLR.cc/2025/Conference — Submitted to ICLR 2025_

### Official Review · Reviewer_GNFh · 2024-10-29

**Soundness:** 2
**Presentation:** 2
**Contribution:** 1
**Rating:** 3
**Confidence:** 5

**Summary:**

This paper introduces a post-hoc explainability method for Graph Neural Networks (GNNs), specifically for the task of graph classification. The authors propose that communities within a graph represent the most crucial subgraphs for understanding the GNN's predictions. By identifying each community and assigning it a score, they aim to pinpoint the most significant subgraph influencing the GNN's decision. The method leverages a well-known community detection algorithm from graph theory to identify these communities. The authors demonstrate the effectiveness of their approach through experiments on various synthetic and real-world datasets.

**Strengths:**

- The problem of explainability in Machine Learning is important and highly relevant.
- The proposed method is simple and straightforward.
- The paper is  easy to read.

**Weaknesses:**

- The authors did not mention or compare their work with the most recent studies in this area (e.g., [1], [2]).
- A significant drawback of this method is its focus on communities. It is possible that only certain nodes within a community are important, but this approach may fail to identify those specific nodes.
- The authors state, "The algorithm is based on the hypothesis that a GNN learns to recognize specific structures in the input graph" and claim that communities are these specific structures. However, they did not formally prove that this is indeed how GNNs learn, nor did they elaborate on or provide evidence for their claims.
- An ablation study using different community detection algorithms is necessary to justify the choice of the one used in their method.
- In Section 3.3, the explanation of datasets is too lengthy and could be partially moved to the appendix. Conversely, the method section is too brief and lacks formal mathematical details.
- The font size in Figure 1 is too small, making it difficult to read.
- The authors should consider incorporating sparsity as a metric and demonstrate how fidelity changes with different subgraph sizes, as the size of the subgraph plays an important role in the quality of the explanation.

[1] Yaochen Xie, Sumeet Katariya, Xianfeng Tang, Edward Huang, Nikhil Rao, Karthik Subbian, and Shuiwang Ji. Task-agnostic graph explanations.
[2] Jialin Chen, Rex Ying. TempME: Towards the Explainability of Temporal Graph Neural Networks via Motif Discovery

**Questions:**

- Could you visualize the subgraphs identified by GECo for different datasets, especially the synthetic ones where the ground truth is known?
- Can you elaborate on the inference process? Specifically, do you need to first identify all communities, feed them individually to the trained GNN, and then select the communities with the highest scores?
- It would be useful if the authors could compare their results with the standard datasets used in previous works (BBBP (Wu et al., 2018), BACE (Wu et al., 2018), and NCI1 (You et al., 2020) are molecular datasets for graph representation learning. BA-Shapes (Yuan et al., 2020; 2021) is a synthetic node classification dataset with 4 unique node labels. The Graph-Twitter (Yuan et al., 2020) dataset is a sentiment graph classification dataset with 3 labels.)

---

### Official Review · Reviewer_a1Dg · 2024-11-01

**Soundness:** 2
**Presentation:** 1
**Contribution:** 1
**Rating:** 1
**Confidence:** 5

**Summary:**

The paper introduces GECo, a new approach designed to improve the explainability of Graph Neural Networks (GNNs) for graph classification tasks by using graph communities. This method works by examining how different communities within a graph contribute to the classification results, generating a mask that highlights the most significant structures in the graph. GECo was assessed on both synthetic and real-world datasets, consistently outperforming existing explainability techniques, such as PGMExplainer, PGExplainer, GNNExplainer, and SubgraphX across different metrics. The findings show that GECo excels at identifying relevant features, satisfying necessity and sufficiency criteria, and aligning with ground-truth explanations. In terms of contribution, this paper  (1) introduces GECo, a novel method for enhancing GNN explainability based on graph communities. (2) demonstrates its great performance and efficiency over previous methods on several datasets.

**Strengths:**

1.	The paper demonstrates that GECo outperforms existing methods in explainability across various datasets. It also highlights GECo's efficiency, offering faster computation times compared to other approaches.

**Weaknesses:**

1. Lack of novelty compared to the conventional graph community method. The proposed explainability method is based on detecting contributory substructures, specifically graph communities. However, the authors do not clearly distinguish the novelty of this approach from conventional graph community-based methods. Since substructure discovery is a widely adopted strategy for explaining GNNs, a more detailed demonstration—either theoretical or empirical—of how the proposed method advances existing graph community approaches is necessary for this work. For example, the paper No.1, Aviyente, Selin, and Abdullah Karaaslanli. "Explainability in Graph Data Science: Interpretability, replicability, and reproducibility of community detection." IEEE Signal Processing Magazine 39.4 (2022): 25-39., and paper No.2 Sangaiah, Arun Kumar, et al. "Explainable AI in big data intelligence of community detection for digitalization e-healthcare services." Applied Soft Computing 136 (2023): 110119. present community-based methods to explain GNNs. A comparison and analysis between your proposed methods and previous ones can be included in this part.

2. Lack of contribution. The abstract and introduction do not clearly articulate the motivation behind the proposed approach or highlight its novelty compared to existing methods. The contribution of using the graph community method for explainability should be well demonstrated in the paper, e.g., how is the proposed method different from the previous studies, and how does this study contribute to the graph community-based explainability methods?

3. Weak and unclear presentation and writing through the paper:
(1) The introduction contains loosely related details and lacks a clear explanation of the background, which may distract readers from the main focus of the work. This work presents a very brief summary of the study in the third paragraph, which doesn’t give readers a clear sense of why using graph communities for explainability is effective. Therefore, a more detailed introduction of the proposed method is needed.
(2) The related work section should be more systematically organized, ideally comparing the strengths and weaknesses of various GNN explainability methods. For example, an organized explainability method for graph explainability is preferred, e.g., following a widely-adopted classification of GNN explainability approaches such as perturbation-based methods, gradient-based, decomposition-based methods, etc., in the related work part will be preferred.
(3) The tables and text on page 8 are not presented professionally or cohesively. More specifically, a more organized arrangement, including both text and table, is preferred.
(4) Section "3.2 The Proposed Methodology" focuses solely on technical details without providing any analysis that would strengthen the theoretical foundation of the proposed approach. For example, providing more details about how the proposed approach detects optimal subgraphs can be helpful for readers to understand the principle behind this method, e.g., mathematical formulation to explain the subgraph detection process.

4. Lack of clear visualization in Fig.1. The workflow in Figure 1 lacks clarity, particularly in how key components, such as the threshold value, are determined. The figure does not clearly illustrate the process, leaving important aspects of the methodology underexplained. A more detailed and explicit visual representation is needed to improve understanding. For example, how to perform graph community in step 2 and how to identify the most influential community can be visualized clearly

5. Lack of sufficient reference. The references cited in this work are insufficient to comprehensively support the proposed methodology. Furthermore, a significant portion of the referenced works are not up-to-date, failing to incorporate the most recent advancements in the field. To strengthen the credibility and relevance of the study, the authors need to include more current and pertinent literature that reflects the latest developments in the area of research. For example, these papers are related to your study: the paper uses graph community for explainability, Aviyente, Selin, and Abdullah Karaaslanli. "Explainability in Graph Data Science: Interpretability, replicability, and reproducibility of community detection." IEEE Signal Processing Magazine 39.4 (2022): 25-39, Sadler, Sophie, Derek Greene, and Daniel Archambault. "Towards explainable community finding." Applied Network Science 7.1 (2022): 81., Martínez Mora, Andrés, et al. "Community-aware explanations in knowledge graphs with XP-GNN." bioRxiv (2024): 2024-01.

**Questions:**

1. Rephrasing and elaborating the introduction section. Try to include more details about the motivation as well as the novelty (e.g., a comparison to the conventional graph communities) of the proposed method. A more detailed and clearer introduction to the proposed method (e.g., the motivation/advantage of using graph community) can be demonstrated in the introduction part.

2. Adding more analysis in the section 3.2. Try to demonstrate not only how it is implemented empirically but why it works theoretically. For example, why and how graph community methods can well detect subgraph for explainability. More nuanced analysis to support the claim is needed.

3. Offering more up-to-date references, especially when introducing the background information of this work. For example, the paper using graph community for explainability, Aviyente, Selin, and Abdullah Karaaslanli. "Explainability in Graph Data Science: Interpretability, replicability, and reproducibility of community detection." IEEE Signal Processing Magazine 39.4 (2022): 25-39.

---

### Official Review · Reviewer_Vv33 · 2024-11-01

**Soundness:** 2
**Presentation:** 3
**Contribution:** 2
**Rating:** 3
**Confidence:** 4

**Summary:**

The paper introduces a method to improve the explainability of GNNs by identifying key communities that contribute to model predictions. GECo uses community detection to isolate important subgraphs, evaluates their impact on the GNN's output, and selects those with high predictive value for explanations.

**Strengths:**

1. The paper is well-organized and clearly written.

2. This paper introduces a novel community-based method for explaining GNNs, focusing on identifying key subgraphs rather than just individual nodes or edges.

**Weaknesses:**

1. GECo uses Blondel et al.'s modularity optimization algorithm for community detection, which performs well on large sparse matrices. However, it does not discuss how different community detection algorithms might impact the explanation results, leading to a lack of robustness verification.

2. GECo determines the threshold 𝜏 by calculating the probability values of communities, using the mean or median as the threshold. However, this method may not be suitable for all cases, especially when the graph structure is uneven or community sizes vary. It is recommended to add experiments exploring adaptive adjustments of 𝜏 in different situations.

3. The baselines compared with GECo are not the latest methods. It would be useful to compare with some instance-level explanation models from the past two years.

4. The paper uses fidelity-based metrics for evaluation. However, these metrics have limitations due to the OOD problem. Therefore, new metrics Fid_{α1,+} and Fid_{α2,-} [1] could be added to assess model fidelity.
[1] Zheng, X., Shirani, F., Wang, T., Cheng, W., Chen, Z.,Chen, H., Wei, H., and Luo, D. Towards robust fidelity for evaluating explainability of graph neural networks. In ICLR, 2024.

**Questions:**

Please see above.

---

### Official Review · Reviewer_i2Vs · 2024-11-04

**Soundness:** 1
**Presentation:** 2
**Contribution:** 1
**Rating:** 1
**Confidence:** 5

**Summary:**

This paper presents GECo, a novel explainability method for Graph Neural Networks (GNNs). GECo focuses on identifying key subgraphs, or communities, that significantly contribute to the model’s classification outcomes. It operates by detecting communities within a graph, then evaluating each subgraph’s individual contribution to the prediction. By setting a probability threshold, GECo isolates the most impactful communities, offering a clear interpretation of the classification results. Tested on synthetic and real datasets, GECo demonstrates superior performance compared to other explainability methods like PGExplainer and GNNExplainer.

**Strengths:**

1. GECo enhances the interpretability of GNNs by identifying the most influential subgraphs, providing a more intuitive and targeted explanation for classification results.

2. GECo achieves significant explainability performance demonstrating effectiveness across both synthetic and real-world datasets.

**Weaknesses:**

1. This paper does not adequately highlight the advantages it has over other explainability models. Specifically, it should analyze the limitations of existing explainability models mentioned in the Related Work section (such as GNNExplainer, PGExplainer, SubgraphX, and PGMExplainer) and convincingly argue the advantages and necessity of a community-based approach for explainability, based on these limitations.

2. The approach in this study is straightforward and lacks novelty. In particular, using communities to generate explanations is an already known method [1]. Additionally, the overall methodology is very similar to studies that recognize motifs and predict their importance to produce explanations [2].

- [1] Martínez Mora, Andrés, et al. "Community-aware explanations in knowledge graphs with XP-GNN." bioRxiv (2024): 2024-01.
- [2] Chen, Jialin, and Rex Ying. "Tempme: Towards the explainability of temporal graph neural networks via motif discovery." Advances in Neural Information Processing Systems 36 (2023): 29005-29028.

3. The proposed model explanation (Section 3.2) is less than one page in length, while the experimental section occupies most of the paper. However, the experimental settings largely replicate those of other studies, leaving little in terms of new insights.

4. Experiments related to runtime performance are necessary.

**Questions:**

The questions are listed in paper weakness.

---

### Meta-Review · Area_Chair_nqa6 · 2024-12-06

**Metareview:**

The paper introduces an algorithm for explaining GNN outcomes using a community-detection-based approach. This method analyzes the contribution of different graph communities to classification results, generating a mask that highlights the most significant structures. GECo was evaluated on synthetic and real-world datasets, outperforming PGMExplainer, PGExplainer, GNNExplainer, and SubgraphX across various metrics.

However, the reviewers point out several limitations in terms of comparison to state-of-the-art explainers, lack of novelty, and unclear presentations. The authors posted a rebuttal, but two reviewers concurred that several of their concerns remain unaddressed. For instance, it was not clear how the choice of metric for community detection affects the explainability metrics. They also noted that the new updates in the rebuttal were not incorporated in the revised manuscript and thus limiting their ability to assess how the work comes together as a whole. Hence, in its current form, the work is not ready for publication.

**Additional Comments On Reviewer Discussion:**

The reviewers remain unanimous in their opinion that, even after considering the rebuttal, the paper has several shortcomings and is thus not ready for publication. The details of the shortcomings are summarized in the meta-review above.

---

### Decision · Program_Chairs · 2025-01-22

Reject